# EFFICIENT PROBABILISTIC TENSOR NETWORKS

## ABSTRACT

Tensor networks (TNs) enable compact representations of large tensors through shared parameters. Their use in probabilistic modeling is particularly appealing, as probabilistic tensor networks (PTNs) allow for tractable computation of marginals. However, existing approaches for learning parameters of PTNs are either computationally demanding and not fully compatible with automatic differentiation frameworks, or numerically unstable. In this work, we demonstrate a conceptually simple approach for learning PTNs efficiently, that is numerically stable. We stabilize the computation of the negative log-likelihood computation by iteratively rescaling intermediate computations using logarithmic scale factors. We show our method provides significant improvements in time and space complexity, achieving 10× reduction in latency for generative modeling on the MNIST dataset. Furthermore, our approach enables learning of distributions with 10× more variables than previous approaches when applied to a variety of density estimation benchmarks. Our code is publicly available at github.com/ptensnet/ptn.

## 1 INTRODUCTION

Generative modeling has seen widespread adoption in recent years, particularly in the areas of language modeling (OpenAI, 2023), image and video generation (Ho et al., 2022), drug discovery (Segler et al., 2018) and material science (Menon & Ranganathan, 2022). These achievements have been made possible by deep neural network based architectures such as Generative Pretrained Transformers (Radford et al., 2018), Generative Adversarial Networks (Goodfellow et al., 2014), Variational Auto-encoders (VAEs) (Kingma & Welling, 2014), Normalizing Flows (Rezende & Mohamed, 2015; Papamakarios et al., 2021) and Diffusion models (Ho et al., 2020).

While generative models used for these applications are remarkably performant in terms of sampling, they fall short in terms of inference. For instance, consider a set of random variables $Y_1, \ldots, Y_N$ and a density $p(Y_1, \ldots, Y_N)$ represented by one of the aforementioned models. Queries such as $p(Y_a|Y_c)$ cannot be performed, where $Y_a, Y_b, Y_c$ are obtained by splitting $(Y_1, \ldots Y_N)$ into three disjoint sets. This is ultimately due to the underlying probabilistic models having intractable marginals (Bond-Taylor et al., 2021).

Tensor networks have been proposed for generative modeling since they allow for tractable marginalization, enabling inference of sophisticated queries such as $p(Y_a|Y_b)$ (Han et al., 2018; Miller et al., 2021a). Motivated by their success in representing many-body quantum states (Schollwöck, 2011b; Orús, 2014), matrix product states (MPS) in particular have been investigated for probabilistic modeling (Han et al., 2018; Vieijra et al., 2022; Glasser et al., 2019). In Glasser et al. (2019), MPS-based models such as Non-Negative Matrix Product States and Born Machines are used for probabilistic modeling. The parameters of these models are learned by minimizing the negative log-likelihood using stochastic gradient descent (SGD). However, this approach does not scale beyond a small number of MPS cores, thereby limiting the number of random variables that can be represented jointly. As shown in Figure 1d, systems with 100 cores or more result in numerical overflows after only two iterations. We analyze this behavior theoretically and show that for Non Negative Matrix product States instability arises due to exponential growth in the magnitude of the expected value of the tensor entries with an increasing number of cores. Meanwhile, for Born Machines it is due to exponential growth in the variance of tensor entries with an increasing number of cores.

Alternatively, Han et al. (2018) and Cheng et al. (2019) use the Density Matrix Renormalization Group (DMRG) algorithm (Schollwöck, 2011b) to learn the model parameters of MPS-based models. While this approach stabilizes the computation of tensor elements due to the isometry of MPS

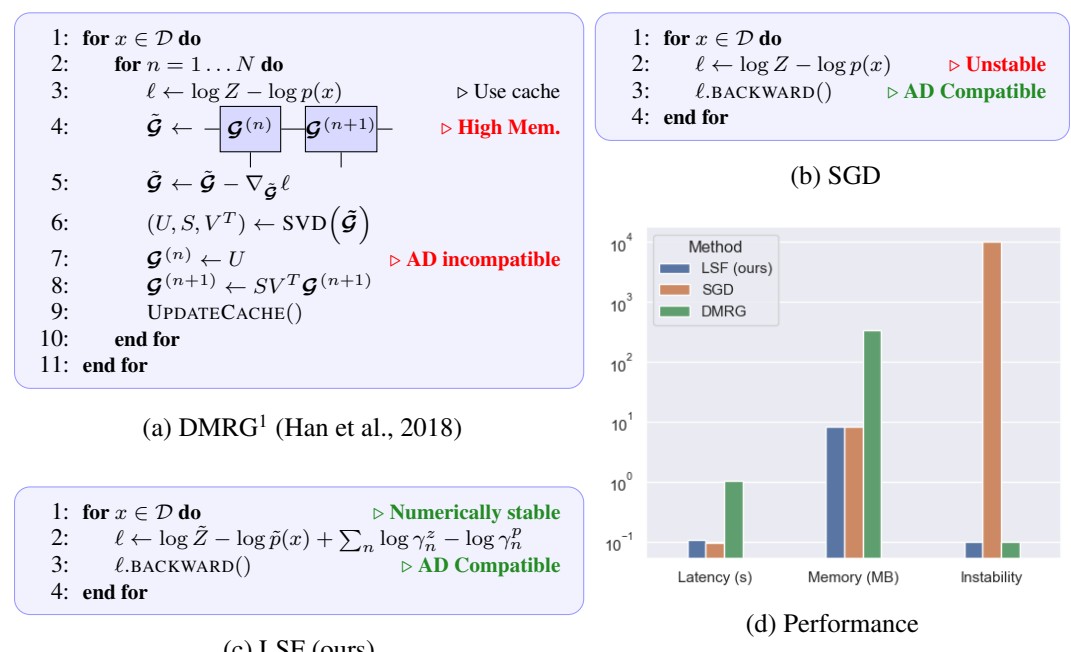

Figure 1: Comparison between training methods for PTNs. (a) DMRG (Han et al., 2018), (b) SGD (Glasser et al., 2019), (c) our method using SGD with logarithmic scale factors (LSF) and (d) latency, memory usage and a measure of instability of the methods. DMRG has exponentially higher latency and memory usage compared with LSF and SGD. However, SGD is numerically unstable. The instability metric is equal to the remaining iterations out of $10^4$ when a numerical overflow is encountered. Even with a modest system size of 100 cores, numerical overflow occurs after just two iterations (see A.1.1 for experimental details).

cores and enables adaptive learning of MPS ranks, it has a number of downsides in practice. First, it is computationally demanding in both space and time, as each parameter update requires performing SVD on a fourth-order tensor as depicted in Figures 1a and 1d. Second, it is not fully compatible with automatic differentiation as the DMRG algorithm does not provide a differentiable loss function that can be used for end-to-end model training as shown in Figure 1a. Third, it is not easily parallelizable across the sequence dimension as updating more than two cores at a time would break the canonical form. While methods for parallelization of DMRG exist, they are even more memory intensive (Stoudenmire & White, 2013). Lastly, we point out that DMRG implementations are non-trivial and require careful maintenance of a cache, which increases the barrier to entry for experimentation with PTNs.

In this work we build off the dynamic rescaling technique proposed in Miller & Rabusseau (2021) for contracting tensor networks using "split formats" and show that our method (i) reliably alleviates the numerical instability issues that arise during the training of PTNs, (ii) achieves significant improvements in both space and time complexity compared with DMRG as shown in Figure 1d and (iii) remains competitive with DMRG on over 20 density estimation benchmarks.

In summary, our contributions are:

- Theoretically analyzing the cause of numerical instability when learning parameters of MPS-based PTNs using stochastic gradient descent and providing lower bounds that characterize the instability of Non Negative Matrix Product states and Born Machines.

- Demonstrating that on real data our method can be used to process sequences that are 10× longer than those managed by previous SGD based approaches. Meanwhile, being 10× faster than alternative numerically stable methods relying on DMRG.

---

[1]This figure uses tensor network diagram notation; see tensornetwork.org/diagrams for reference.

## 2 RELATED WORK

**Tensor Networks** have been used to approximate high dimensional tensors in a variety of domains including neuroscience (Williams et al., 2018; Cong et al., 2015), chemistry (Murphy et al., 2013) and hyperspectral imaging (Fang et al., 2017). Classical learning methods include the Higher-Order Singular Value Decomposition (HOSVD) for Tucker models (Kolda & Bader, 2009), Alternating Least Squares (ALS) for Canonical Polyadic (CP) decompositions (Carroll & Chang, 1970), and the Tensor-Train SVD (TT-SVD) for tensor-train (TT) formats (Oseledets, 2011). In quantum many-body physics, the Density Matrix Renormalization Group (DMRG) provides a powerful scheme for optimizing matrix product states (MPS) through local updates (Schollwöck, 2011a), enabling adaptive bond dimensions.

**Probabilistic modeling with Tensor Networks (TNs)** has been investigated for uni-variate conditional distributions (Novikov et al., 2017; Stoudenmire & Schwab, 2016), multi-variate distributions (Han et al., 2018; Novikov et al., 2021; Cheng et al., 2019; Glasser et al., 2019), as well as sequence modeling tasks (Miller et al., 2021a). Previous work for training probabilistic models in the context of density estimation (Novikov et al., 2021; Glasser et al., 2019; Wu et al., 2023; Ghalamkari et al., 2024) has been limited to small-scale settings ($< \mathbf{70}$ **random variables**) or relied on DMRG (Han et al., 2018; Cheng et al., 2019) for optimization. In contrast, we make use of SGD and stabilize the computation using logarithmic scale factors, showing that we can fit distributions with upwards of 10,000 random variables. Lastly, our approach builds on the dynamic rescaling method presented in Miller & Rabusseau (2021), which proposes to track a single multiplicative scaling factor throughout a tensor network contraction. While the approach of Miller & Rabusseau (2021) is more general, our method leverages the structure of the negative log-likelihood objective to track a series of differences of multiplicative scale factors.

**Relationships between PTNs and alternative probabilistic modeling frameworks** such as Probabilistic Graphical Models (PGMs) and Probabilistic Circuits (PCs) have been previously investigated. In Glasser et al. (2019), mappings between hidden markov models and non negative MPS-based distributions have been provided, as well as mappings between quantum circuits and MPS-based born machines. Furthermore, in Loconte et al. (2025), the Tucker decomposition and MPS have been shown to have equivalent shallow and deep probabilistic circuit representations, respectively. Moreover both Loconte et al. (2025) and Ciolli et al. (2025) fit Non-Negative MPS models on high dimensional problems, relying on the stability of probabilistic circuit frameworks. In contrast, our work formally characterizes the instability issue that arises when training Born Machines in particular, in addition to Non-Negative MPS. Lastly, Miller et al. (2021b) provides a hybrid framework for PGMs and PTNs.

## 3 METHOD

We consider the task of modeling multi-variate distributions of the form

$$p(y_1, y_2, \ldots, y_N), \tag{1}$$

where $y_i \in \mathcal{Y}_N$ is a discrete random variable. Since a direct representation of Equation 1 is generally intractable and learning in such high-dimensional spaces is hindered by the curse of dimensionality, one typically resorts to parametric approaches. Matrix Product States is a class of parametric models that can represent such distributions with the added benefit that marginals are tractable to compute.

The rest of this section is organized as follows: Section 3.1 introduces the Matrix Product State (MPS) model. Sections 3.2 & 3.3 define MPS-based probabilistic tensor networks (PTNs). Section 3.4 outlines the trade-offs between using DMRG and SGD to train MPS-based models and provides a theoretical analysis of the stability issue encountered when using SGD to train PTNs. Section 3.5 introduces our method using logarithmic scale factors. Section 3.6 compares our proposed method with the Density Matrix Renormalization Group (DMRG) for learning MPS-based models, in terms of compatibility with automatic differentiation. Lastly, Section 3.7 demonstrates how sampling can be performed using MPS-based probabilistic models.

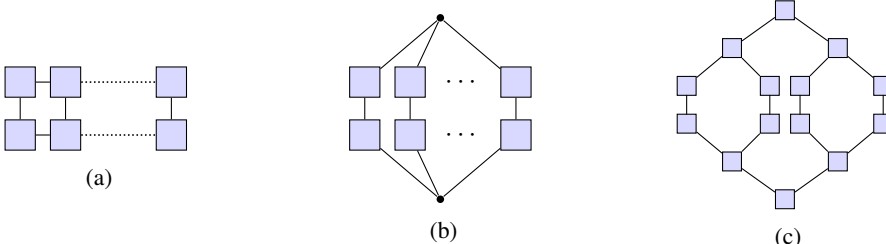

Figure 2: Normalization constant of various PTNs. (a) MPS, (b) CP, and (c) Tensor Tree

## 3.1 MATRIX PRODUCT STATES

The Matrix Product State (MPS) model provides a structured representation of high-order tensors by factorizing them into a sequence of matrices. Thus, tensor $\mathcal{T} \in \mathbb{R}^{D_1 \times D_2 \times \cdots \times D_N}$ can be approximated as a sequence of matrix multiplications

$$\mathcal{T}_{y_1 \cdots y_h} \approx \mathcal{G}^{(1)}[y_1] \cdots \mathcal{G}^{(N)}[y_N] \tag{2}$$

where $\mathcal{G}^{(i)} \in \mathbb{R}^{R_i \times D_i \times R_{i+1}}$ are referred to as the MPS *cores*, $R_i$ is referred to as the *Ranks* or *Bond Dimensions*, $D_i$ are referred to as the *input dimensions*, $[\cdot]$ indicates slicing along the input dimension (i.e., $\mathcal{G}^{(i)}[k] \in \mathbb{R}^{R_i \times R_{i+1}}$), and the boundaries are constrained such that $R_i = r_N = 1$, making the contraction in Equation 2 scalar valued. Assuming equal dimensions $D_i = D$ for all $i$, the MPS parameterization has a space complexity of $\mathcal{O}(NDR^2)$ which is *linear* in $N$, compared with $\mathcal{O}(D^N)$ in the original tensor.

## 3.2 PROBABILISTIC MODELING WITH MPS$_{\text{BM}}$

In order to represent a valid probability distribution using the MPS model, the parameters must be constrained such that all the tensor entries are positive and sum to one. Born Machines enforce such constraints by taking inspiration from quantum mechanics where wavefunctions induce probability distributions described by the squared norm of the wavefunction

$$p(y_1, \ldots, y_N) = \frac{|\Psi_{\text{BM}}(\mathbf{y})|^2}{Z}, \quad \Psi_{\text{BM}}(\mathbf{y}) = \mathcal{G}^{(1)}[y_1] \cdots \mathcal{G}^{(N)}[y_N] \tag{3}$$

where $y_i \in \mathcal{Y}_i \subset \mathbb{N}$ and $\mathbf{y} = (y_1, \ldots, y_N)$. At first glance, it may seem that computing the normalization constant $Z$ in Equation 3 requires a summation over an exponential number of terms, as it would require summing up all the squares of the elements in the underlying tensor

$$Z = \sum_{y_1', \ldots y_N' \in \mathcal{Y}^N} \left( \mathcal{G}^{(1)}[y_1'] \cdots \mathcal{G}^{(N)}[y_N'] \right)^2. \tag{4}$$

Remarkably, a key property of the underlying MPS model is the ability to compute the normalization constant efficiently, in time linear in $N$. Algebraically, the computation simplifies to

$$Z = \sum_{r_1, r_2, y_1'} \mathcal{G}^{(1)}_{r_1, r_2}[y_1'] \mathcal{G}^{(1)}_{r_1, r_2}[y_1'] \quad \cdots \quad \sum_{r_N, r_{N+1}, y_N'} \mathcal{G}^{(N)}_{r_N, r_{N+1}}[y_N'] \mathcal{G}^{(N)}_{r_N, r_{N+1}}[y_N'].$$

This property is easy to see using tensor network diagrams as depicted in Figure 2a. Notably, this property is not unique to MPS and other tensor network structures such as Canonical Polyadic (CP) and Tensor Tree exhibit similar simplifications as shown in Cheng et al. (2019).

## 3.3 PROBABILISTIC MODELING WITH MPS$_\sigma$

We now introduce the MPS$_\sigma$ model, which enforces positivity of the underlying tensor by enforcing positivity on each of the cores independently. In other words,

$$p(y_1, \ldots, y_N) = \frac{\Psi_\sigma(\mathbf{y})}{Z}, \quad \Psi_\sigma(\mathbf{y}) = \sigma(\mathcal{G}^{(1)})[y_1] \cdots \sigma(\mathcal{G}^{(N)})[y_N] \tag{5}$$

where $\sigma : \mathbb{R} \to \mathbb{R}_{\geq 0}$ is a non-negative function, applied point-wise to tensor entries. The normalization constant in Equation 5 can be computed efficiently and reduces to a sequence of matrix multiplications (see Appendix A.3).

### 3.4 LEARNING MPS$_{\text{BM}}$ AND MPS$_\sigma$

Previous methods have shown that DMRG can be used to learn the parameters of MPS$_{\text{BM}}$ (Han et al., 2018), and SGD can be used to learn the parameters of both MPS$_{\text{BM}}$ and MPS$_\sigma$ (Glasser et al., 2019) by minimizing the negative log likelihood. However, both approaches have shortcomings that we summarize in Table 1. While DMRG is numerically stable, it is computationally intensive (see Table 2) and not fully compatible with automatic differentiation, thereby making it difficult to integrate into machine learning frameworks (see Section 3.6). We also note that the combination of DMRG and MPS$_\sigma$ is not well defined. This is because naively applying a non-linearity after the decomposition would corrupt the parameter update (see Appendix A.4). This incompatibility further restricts the applicability of DMRG for training PTNs. Given the downsides of using DMRG, we revisit using vanilla SGD as in Glasser et al. (2019).

**Why can we not use vanilla SGD to train MPS$_\sigma$ models?**

The MPS$_\sigma$ model enforces positivity of the underlying tensor by applying a point-wise positivity function to each of the MPS cores as in Equation 5. However, this constraint causes both the numerator and denominator in Equation 5 to grow rapidly with the number of cores as shown in Figure 3. We characterize this growth in Theorem 1, showing that the expected value of both the numerator and denominator grow *exponentially* with the rank dimension (proof in Appendix A.7)

**Theorem 1.** *Let the elements of the tensor $\boldsymbol{\mathcal{G}}^{(i)} \in \mathbb{R}^{R_i \times D \times R_{i+1}}$ be i.i.d. random variables drawn from a zero-mean gaussian distribution with unit variance, $R_1 = R_N = 1$ and $R_i = R \quad \forall i \neq 1, N$. Let $\mathbf{y} \in \mathcal{Y}$, $\mathcal{Y} = \mathcal{Y}_i \times \cdots \times \mathcal{Y}_N$ and*

$$\Psi_\sigma(\mathbf{y}) = \sigma(\boldsymbol{\mathcal{G}}^{(1)}[y_1]) \cdots \sigma(\boldsymbol{\mathcal{G}}^{(N)}[y_N]), \quad Z_\sigma = \dot{\boldsymbol{\mathcal{G}}}^{(1)} \cdots \dot{\boldsymbol{\mathcal{G}}}^{(N)}, \tag{6}$$

*where $\dot{\boldsymbol{\mathcal{G}}}_{ij} \triangleq \sum_k \sigma(\boldsymbol{\mathcal{G}}_{ikj})$ and $\sigma : \mathbb{R} \to \mathbb{R}_{\geq 0}$ is a point-wise non-negative mapping s.t.*

$$\forall x \in \mathbb{R}_{>0}, \ \exists \epsilon_x > 0 \ s.t. \ \sigma(x) > \epsilon_x.$$

*Then, $\mathbb{E}[\Psi_\sigma(\mathbf{y})] \geq \epsilon R^N$ and $\mathbb{E}[Z_\sigma] \geq \epsilon R^N D^N$ for some $\epsilon > 0$*

**Why can we not use vanilla SGD to train MPS$_{\text{BM}}$ models?**

In contrast to MPS$_\sigma$, MPS$_{\text{BM}}$ does not enforce positivity on cores and does not suffer as drastic a growth in magnitude, since cancellation can occur between negative and positive terms in the tensor contraction as shown in Figure 3. However, while the expected value of $\Psi_{\text{BM}}(\mathbf{y})$ in Equation 3 is zero (with respect to cores $\boldsymbol{\mathcal{G}}$), its *variance* grows exponentially with the rank dimension $R$, as shown in Theorem 2 (proof in Appendix A.7)

**Theorem 2.** *Let the elements of the tensor $\boldsymbol{\mathcal{G}}^{(i)} \in \mathbb{R}^{R_i \times D \times R_{i+1}}$ be i.i.d. random variables drawn from a zero-mean gaussian distribution with unit variance, $R_1 = R_N = 1$ and $R_i = R \quad \forall i \neq 1, N$. Let $\mathbf{y} \in \mathcal{Y}$, $\mathcal{Y} = \mathcal{Y}_i \times \cdots \times \mathcal{Y}_N$ and $\Psi_{\text{BM}}(\mathbf{y}) = \boldsymbol{\mathcal{G}}^{(1)}[y_1] \cdots \boldsymbol{\mathcal{G}}^{(N)}[y_N]$. Then, $\mathbb{E}[\Psi_{\text{BM}}(\mathbf{y})] = 0$ and $\mathbb{E}[\Psi_{\text{BM}}(\mathbf{y})^2] \geq \epsilon R^N$ for some $\epsilon > 0$*

Overall, the instability of using vanilla SGD for training PTNs severely limits it's applicability to real world datasets. For instance, in Glasser et al. (2019) only datasets consisting of a maximum of 22 variables were considered.

Table 1: Trade-offs between different combinations of models and optimization routines (ME = Memory Efficient; PLL = Parallelizable; AD = compatible with Automatic Differentiation). Notably, LSF (ours) outperforms SGD and DMRG in terms of stability and computational intensity, respectively.

| Optimization | Model | Adaptive | Fast | ME | PLL | AD | Stable |
|---|---|---|---|---|---|---|---|
| DMRG (Han et al., 2018) | MPS$_{\text{BM}}$ | ✓ | ✗ | ✗ | ✗ | ✗ | ✓ |
| SGD (Glasser et al., 2019) | MPS$_{\text{BM}}$ | ✗ | ✓ | ✓ | ✓ | ✓ | ✗ |
| SGD (Glasser et al., 2019) | MPS$_\sigma$ | ✗ | ✓ | ✓ | ✓ | ✓ | ✗ |
| LSF (ours) | MPS$_{\text{BM}}$ | ✗ | ✓ | ✓ | ✓ | ✓ | ✓ |
| LSF (ours) | MPS$_\sigma$ | ✗ | ✓ | ✓ | ✓ | ✓ | ✓ |

Table 2: Asymptotic time and space complexities of DMRG vs SGD.

| Complexity | Expression |
|---|---|
| **Time** | |
| $\text{MPS}_{\text{BM+DMRG}}$ | $\mathcal{O}(\text{NR}^3\text{D} + \text{NR}^2\text{D}^2)$ |
| $\text{MPS}_{\sigma+\text{SGD}}$ | $\mathcal{O}(\text{NR}^3 + \text{ND})$ |
| **Space** | |
| $\text{MPS}_{\text{BM+DMRG}}$ | $\mathcal{O}(\text{NDR}^2 + \text{D}^2\text{R}^2)$ |
| $\text{MPS}_{\sigma+\text{SGD}}$ | $\mathcal{O}(\text{NDR}^2)$ |

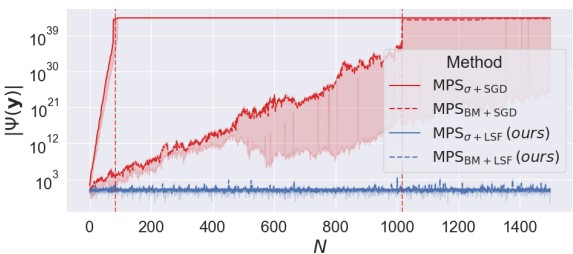

Figure 3: Magnitude of numerator terms in and Equations 3 and 5 as $N$ is increased for MPS-based (See Appendix A.1.2 for experimental details).

---

**Algorithm 1** LSF (Stochastic Gradient Descent with Logarithmic Scale Factors)

---

**Require:** Data $Y \in \mathbb{N}^{N_{\text{samples}} \times D}$, parameters $g_i \in \mathbb{R}^{R_i \times D_i \times R_{i+1}}$
**Ensure:** Updated parameters $\{g_i^\star\}_{i=1}^N$
1: **for** $i = 1 \ldots N_{\text{samples}}$ **do**
2:     $\tilde{p}_1 \leftarrow \mathbf{1}_1, z_1 \leftarrow \mathbf{1}_1$                                       ▷ One dimensional vector
3:     $\gamma_1^{(p)} \leftarrow 1, \gamma_1^{(z)} \leftarrow 1$
4:     $\tilde{p}_n \leftarrow \frac{1}{\gamma_n^{(p)}}\tilde{p}_{n-1}^T\sigma(\boldsymbol{\mathcal{G}}^{(n-1)}[Y_{in}])$
5:     $\gamma_n^{(z)} \leftarrow \max(\tilde{p}_{n-1}^T\sigma(\boldsymbol{\mathcal{G}}^{(n-1)}[Y_{in}]))$
6:     $z_n \leftarrow \frac{1}{\gamma_n^{(z)}}z_{n-1}^T\dot{\boldsymbol{\mathcal{G}}}^{(n-1)}[Y_{in}]$
7:     $l \leftarrow (\log\sum_n z_{N+1} - \tilde{p}_{N+1}) + \sum_j \log\gamma_j^{(z)} - \log\gamma_j^{(p)}$
8:     $\theta \leftarrow \theta - \alpha\nabla_\theta l$
9: **end for**
10: **return** $\theta$

---

### 3.5 SGD WITH LOGARITHMIC SCALE FACTORS

This section introduces our numerically stable method for computing the negative log-likelihood of MPS-based probabilistic tensor networks. We demonstrate our method on the $\text{MPS}_{\text{BM}}$ class of models and show that it can be straightforwardly extended to the $\text{MPS}_\sigma$ class. Given a dataset of $K$ independent and identically distributed (i.i.d.) observations $\mathcal{D} = \{\mathbf{y}_N^{(k)}\}_{k=1}^K$, we learn the model parameters $\boldsymbol{\mathcal{G}}^{(i)}$ of an $\text{MPS}_{\text{BM}}$ distribution using maximum likelihood estimation, by minimizing

$$\ell(\boldsymbol{\mathcal{G}}) = -\frac{1}{K}\sum_{k=1}^K \log Z - 2\log\left|\Psi_{\text{BM}}(\mathbf{y}^{(k)})\right|. \tag{7}$$

Computing the sequence of matrix multiplications in Equation 3 leads to a numerical overflow as shown in Figure 5. Since we are ultimately concerned with computing log probabilities, we can factor out ***a product of logarithms of scale factors*** in order to stabilize the computation. Equivalently, the loss can be written as:

$$\ell(\boldsymbol{\mathcal{G}}) = \log\frac{Z}{\Sigma_i\gamma_i^{(z)}} - 2\log\frac{\Psi_{\text{BM}}(\mathbf{y})}{\Sigma_i\gamma_i^{(p)}} + \sum_n \log\gamma_n^{(z)} - \log\gamma_n^{(p)}. \tag{8}$$

This raises the question of how to select the scaling factors $\gamma_i^{(z)}$ and $\gamma_i^{(p)}$ in a manner that guarantees numerical stability throughout all intermediate computations. We address this by computing these stabilization factors *iteratively*. Let

$$\tilde{\Psi}_{\text{BM}}(\mathbf{y}) = \frac{\Psi_{\text{BM}}(\mathbf{y})}{\Sigma_i\gamma_i^{(p)}} = \widetilde{\boldsymbol{\mathcal{G}}}^{(1)}[y_1]\cdots\widetilde{\boldsymbol{\mathcal{G}}}^{(N)}[y_N]$$

where intermediate tensors $\widetilde{\boldsymbol{\mathcal{G}}}^{(n)}$ and scale factors $\gamma_i^{(p)}$ are given by

$$\widetilde{\boldsymbol{\mathcal{G}}}^{(n)} := \frac{1}{\gamma_n^{(p)}}\widehat{\boldsymbol{\mathcal{G}}}^{(n)}, \quad \gamma_n^{(p)} := \left\|\widehat{\boldsymbol{\mathcal{G}}}^{(n)}\right\|, \quad \widehat{\boldsymbol{\mathcal{G}}}^{(n)} := \widetilde{\boldsymbol{\mathcal{G}}}^{(1)}[y_1]\cdots\widetilde{\boldsymbol{\mathcal{G}}}^{(n-1)}[y_{n-1}]\sigma(\boldsymbol{\mathcal{G}}^{(n)}[y_n]). \tag{9}$$

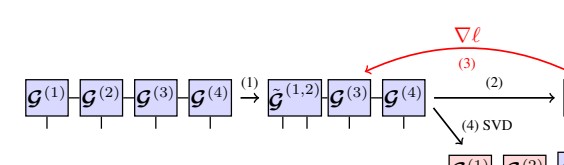

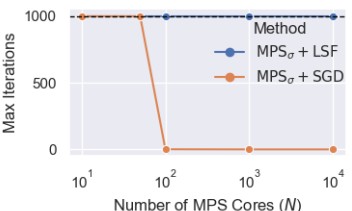

Figure 4: Illustration of a single update step using the DMRG two site update algorithm used in Han et al. (2018). (1) Cores $\mathcal{G}^{(1)}$ and $\mathcal{G}^{(2)}$ are merged, (2) the loss is computed with respect to the merged fourth order tensor, (3) the gradient is computed and used to update the fourth order tensor using automatic differentiation and (4) the fourth order tensor is decomposed using SVD, then singular vectors are *copied* into cores $\mathcal{G}^{(1)}$ and $\mathcal{G}^{(2)}$.

Figure 5: Maximum number of iterations reached during training using vanilla stochastic gradient descent $\mathrm{MPS}_{\sigma+\mathrm{SGD}}$ vs. stochastic gradient descent with logarithmic scale factors $\mathrm{MPS}_{\sigma+\mathrm{LSF}}$ (See Appendix A.1.3 for details).

The same progression can be used to stabilize the computation of the normalization constant and the overall procedure can be straightforwardly extended to the $\mathrm{MPS}_{\sigma}$ class.

Finally, we highlight that when the $\mathrm{MPS}_{\sigma}$ family is restricted to the exponential activation for enforcing positivity, our procedure reduces to an instance of iterative scaling based on the well-known Log-Sum-Exp stabilization similar to the approach used in Peharz et al. (2020). Importantly, this correspondence does not extend to Born Machines, for which the Log-Sum-Exp trick is not directly applicable (See Appendix A.6 for more details).

### 3.6 COMPATIBILITY WITH AUTOMATIC DIFFERENTIATION

The method proposed in Section 3.5 enables the stable computation of the negative log-likelihood using Equation 8, thus end-to-end learning of PTN model parameters can be performed. In contrast, the DMRG algorithm uses the negative log likelihood to compute updates with respect to a fourth-order tensor that is subsequently decomposed using SVD. This decomposition serves two purposes: (i) it enables adaptive learning of bond dimensions and (ii) it maintains isometry of cores, which in turn stabilizes the computation of the loss.

A single step of the DMRG algorithm used in Han et al. (2018) is depicted in Figure 4. First, the neighboring cores $\mathcal{G}^{(1)}$ and $\mathcal{G}^{(2)}$ are merged. Second, the loss function (negative log likelihood) is computed. Third, the gradient of the loss function is computed with respect to the fourth-order tensor (this can be done using automatic differentiation) and used to update the fourth order tensor using gradient descent. Lastly, the updated the fourth order tensor is decomposed using SVD and singular vectors are copied into the model parameters $\mathcal{G}^{(1)}, \mathcal{G}^{(2)}$. Crucially, the last step is not an ancestor of the loss function computation, thus model parameters cannot be updated end-to-end using automatic differentiation.

### 3.7 SAMPLING FROM MPS-BASED MODELS

Sampling from MPS-based models can de done efficiently and reduces to performing a sequence of matrix multiplications. Conditional sampling can also be performed efficiently, due to the tractable computation of marginals. For instance, in order to sample in an auto-regressive fashion, we can compute the conditional distribution for the $n^{\text{th}}$ position given the past as follows:

$$p(y_n \mid y_1, \ldots, y_{n-1}) = \frac{p(y_1, \ldots, y_n)}{p(y_1, \ldots, y_{n-1})} = \frac{\mathcal{G}^{(1)}[y_1] \cdots \mathcal{G}^{(n)}[y_n] \dot{\mathcal{G}}^{(n+1)}[y_{n+1}] \cdots \dot{\mathcal{G}}^{(N)}[y_N]}{Z},$$

where $\dot{\mathcal{G}}_{ij} = \sum_k \mathcal{G}_{ikj}$. Notably, with MPS-based models we can sample in any order and from any marginal distribution (see Appendix A.5).

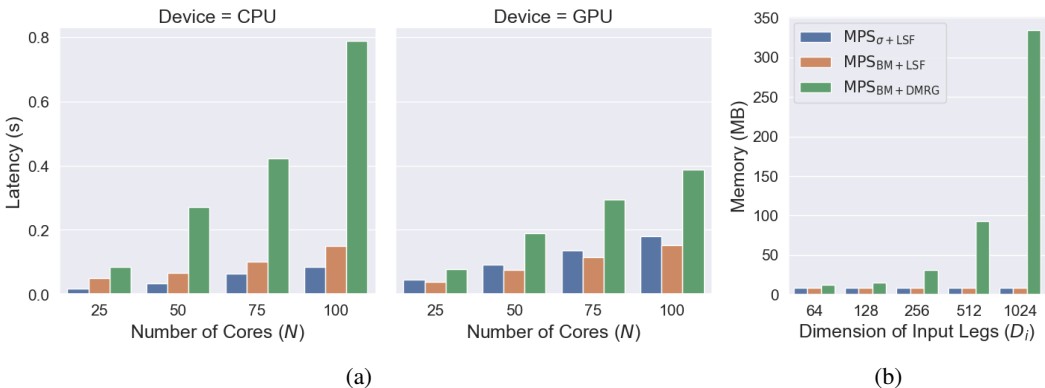

(a)

(b)

Figure 6: (a) Latency of single update to all model parameters using LSF and DMRG, on both CPU and GPU for various number of cores ($N$). (b) Peak memory encountered during a single update to all model parameters using LSF and DMRG for various free leg dimensions.

# 4 EXPERIMENTS

In this section we compare our method LSF with both vanilla SGD and DMRG for training different MPS-based probabilistic tensor networks. Section 4.1 compares the stability of LSF vs. SGD. Section 4.2 compares latency and memory requirements of LSF vs. DMRG for varying MPS model dimensions. Section 4.3 compares the performance of LSF vs. SGD on various density estimation benchmarks and Section 4.4 compares the performance of LSF vs. DMRG on MNIST.

Table 3: Average test negative log-likelihood achieved by different tensor network models trained on MNIST.

| Model | NLL | Lat (s) |
|---|---|---|
| PixelCNN | 0.104 | - |
| MPS$_{BM+DMRG}$ | 0.129 | 1.20 |
| MPS$_{exp+LSF}$ (ours) | 0.136 | 0.11 |
| MPS$_{abs+LSF}$ (ours) | 0.140 | 0.12 |
| MPS$_{sig+LSF}$ (ours) | 0.148 | 0.12 |
| MPS$_{BM+LSF}$ (ours) | 0.168 | 0.12 |

## 4.1 COMPARING THE STABILITY OF LSF VS. SGD

This section analyzes the numerical stability of training MPS-based models using stochastic gradient descent with logarithmic scale factors (LSF) vs. vanilla stochastic gradient descent (SGD) as in Glasser et al. (2019). In Figure 5 we train MPS$_\sigma$ for up to 1k iterations while varying the number of cores. For smaller systems (approximately $N \approx 50$), SGD successfully updates the MPS cores. However, for systems exceeding $N = 100$, numerical overflow prevents more than a single iteration to be performed. In contrast, LSF enables training for the maximum number of iterations even with 10k MPS cores.

Table 4: Average test negative log-likelihood for LSF compared to SGD (Glasser et al., 2019) and EiNet (EN) (Peharz et al., 2020) methods. The † symbol indicates numerical overflow occurred before training completion, while ✗ indicates numerical overflow before completing a single epoch (see Appendix A.2 for experimental details).

| Data | N | EiN | SGD | DMRG | LSF |
|---|---|---|---|---|---|
| nltcs | 16 | 0.38 | 0.38 | 0.38 | 0.38 |
| msnbc | 17 | 0.35 | 0.36 | 0.36 | 0.36 |
| kdd | 64 | 0.03 | 0.33 | 0.03 | 0.03 |
| plants | 69 | 0.2 | 0.37 | **0.20** | 0.24 |
| jester | 100 | 0.53 | ✗ | 0.55 | **0.54** |
| baudio | 100 | 0.4 | ✗ | 0.43 | **0.42** |
| bnetflix | 100 | 0.57 | ✗ | 0.61 | **0.59** |
| accidents | 111 | 0.34 | ✗ | **0.33** | 0.35 |
| retail | 135 | 0.08 | ✗ | 0.08 | 0.08 |
| pbstar | 163 | 0.24 | ✗ | **0.19** | 0.23 |
| dna | 180 | 0.54 | ✗ | 0.46 | **0.44** |
| kosarek | 190 | 0.06 | ✗ | 0.06 | 0.06 |
| msweb | 294 | 0.04 | ✗ | **0.03** | 0.04 |
| book | 500 | 0.07 | ✗ | 0.07 | 0.07 |
| tmovie | 500 | 0.11 | ✗ | 0.12 | 0.12 |
| cwebkb | 839 | 0.19 | ✗ | 0.21 | **0.20** |
| cr52 | 889 | 0.1 | ✗ | 0.11 | 0.11 |
| c20ng | 910 | 0.17 | ✗ | ✗ | **0.18** |
| bbc | 1058 | 0.25 | ✗ | ✗ | **0.26** |
| ad | 1556 | 0.04 | ✗ | ✗ | 0.04 |

## 4.2 LATENCY AND MEMORY USAGE OF LSF VS. DMRG

We analyze the latency and peak memory usage of LSF compared with DMRG in Figure 6 on both CPU and GPU. We set the batch size, ranks $R_i$ and free legs $D_i$ to 32, 8 and 2 respectively. Meanwhile, we vary the number of cores $N$. As shown in Figure 6a, $\text{MPS}_{\sigma+\text{LSF}}$ and $\text{MPS}_{\text{BM}+\text{LSF}}$ achieve drastic speedups over $\text{MPS}_{\text{BM}+\text{DMRG}}$ (Han et al., 2018) as the number of cores $N$ is increased. For instance, with $N = 100$, $\text{MPS}_{\text{BM}+\text{DMRG}}$ requires **0.8** seconds to perform one update to all model parameters; meanwhile, $\text{MPS}_{\sigma+\text{LSF}}$ requires **0.09** seconds, leading to approximately one order of magnitude speedup.

We compare the peak memory usage of both methods in Figure 6b at various input dimensions. As illustrated in Figure 1, $\text{MPS}_{\text{BM}+\text{DMRG}}$ requires the materialization of fourth-order tensors during training. We show in Figure 6b that this quickly leads to extreme memory consumption. For instance, at $D_i = 1024$ $\text{MPS}_{\text{BM}+\text{DMRG}}$ requires **334 MB** compared with only **8 MB** for $\text{MPS}_{\sigma+\text{LSF}}$.

## 4.3 PERFORMANCE OF LSF VS. SGD ON DENSITY ESTIMATION BENCHMARKS

This section compares the generalization performance of MPS-based models trained using SGD (Glasser et al., 2019) vs. LSF on 20 density estimation benchmarks (Lowd & Davis, 2010; Van Haaren & Davis, 2012). We also compare against EiNet, a state-of-the-art probabilistic circuit with tractable marginals (Peharz et al., 2020). Table 4 reports the best test set performance. The ✗ symbol indicates numerical overflow before completing the first epoch. Notably, our method achieves comparable performance with EiNet, meanwhile the approach in Glasser et al. (2019) results in numerical overflow on most datasets. Specifically, **SGD fails entirely on all datasets with 100 random variables or more** and partially on datasets consisting of $\sim 60$ random variables.

## 4.4 COMPARING MNIST GENERALIZATION PERFORMANCE OF LSF VS. DMRG

This section compares the generalization performance of MPS-based models trained with DMRG vs. LSF on the task of learning to generate MNIST digits. We use 60K and 10K samples for training and testing, respectively. Each image is flattened and binarized to produce a 784-dimensional binary vector. We then train MPS models using LSF, setting $N = 784$, $D_i = 2$, and $R_i = 32$. Table 3 demonstrates that $\text{MPS}_{\sigma+\text{LSF}}$ achieves performance comparable to $\text{MPS}_{\text{BM}+\text{DMRG}}$ while providing approximately **10×** **speedup**. The memory advantages of LSF over DMRG are negligible in this experiment as DMRG's memory usage scales quadratically with input dimensions $D_i$, which only equals two in this experiment. We also benchmark against PixelCNN, which achieves state-of-the-art performance on this task. Although PixelCNN outperforms both MPS approaches, it lacks tractable marginals, thus cannot be used for inference of complex queries.

Since LSF enables training a wider range of MPS-based models than DMRG, we experiment with various positivity enforcing functions. We find that $\text{MPS}_{\sigma}$ models generally outperform $\text{MPS}_{\text{BM}}$ counterparts using LSF.

## 5 CONCLUSION

Probabilistic Tensor Networks (PTNs) enable tractable inference over high-dimensional distributions, but face significant training challenges. Previous work has been limited to small-scale experiments ($< 50$ variables) or relied on the computationally intensive DMRG algorithm for stable learning of a particular subset of PTNs. Beyond its computational cost, the reliance on DMRG presents a significant barrier to experimentation with PTNs, as DMRG implementations require non-trivial cache management for efficient batch processing, and do not leverage automatic differentiation for end-to-end model training (Zhang, 2018).

In this work, we addressed these limitations by introducing a stable method for the computation of the negative log-likelihood based on logarithmic scale factors. While our method was focused on MPS-based PTNs, as a future work it can be extended to other tensor networks structures. This approach enables larger scale training of PTNs, making them more practical for real-world applications. These advances also enable experimentation with PTNs using standard deep learning pipelines, while also opening exploration of the broad $\text{MPS}_{\sigma}$ class of PTNs.

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

# A APPENDIX

## A.1 EXPERIMENTAL DETAILS

### A.1.1 EXPERIMENTAL DETAILS FOR FIGURE 1D

In this experiment we use the hyper-parameters listed in Table 5. The instability metric is computed using the following equation

$$\text{Instability} = \text{Max Iterations Reached} - 10000 + 0.1,$$

where the maximum number of iterations possible is 10k.

Table 5: Hyper-parameters for experiments shown in Figure 1d.

| HP | Latency | Instability | Memory |
|---|---|---|---|
| Batch Size | 32 | 32 | 32 |
| Rank | 2 | 2 | 2 |
| Input leg | 2 | 2 | 1024 |
| Number of cores | 100 | 100 | 5 |

### A.1.2 EXPERIMENTAL DETAILS FOR FIGURE 3

In this experiment we compare the loss computation associated with performing stochastic gradient descent (SGD) compared with the loss computation (forward pass) associated with our method using logarithmic scale factors (LSF). Importantly, no backward passes are performed.

### A.1.3 EXPERIMENTAL DETAILS FOR FIGURE 5

In this experiment we use LSF with a learning rate of $1e-3$ and for SGD we report the maximum iterations achievable after trying both vanilla SGD and Adam with learning rates $\{$1e-3, 5e-4, 1e-4, 5e-5, 1e-5, 1e-6$\}$.

## A.2 EXPERIMENTAL DETAILS FOR FIGURE 4

In this experiment, we use Adam as the SGD baseline as we find it achieves better performance. We train all models for 50 epochs with batch size 32, bond dimension of 32, learning rate of 5e-3. We train $\text{MPS}_\sigma$ models using LSF and select the exponential function for positivity.

## A.3 COMPUTING THE NORMALIZATION CONSTANT OF $\text{MPS}_\sigma$

The normalization constant $Z$ of the $\text{MPS}_\sigma$ class can be computed in time linear in $N$, following a similar algebraic simplification as shown for the Born Machine in Equation 4,

$$Z = \sum_{y_1', \dots y_N' \in \mathcal{Y}^N} \sigma\left(\boldsymbol{\mathcal{G}}^{(1)}[y_1']\right) \cdots \sigma\left(\boldsymbol{\mathcal{G}}^{(N)}[y_N']\right) \tag{10}$$

$$= \sum_{r_1, r_2, y_1'} \sigma\left(\boldsymbol{\mathcal{G}}^{(1)}_{r_1, r_2}[y_1']\right) \cdots \sum_{r_N, r_{N+1}, y_N'} \sigma\left(\boldsymbol{\mathcal{G}}^{(N)}_{r_N, r_{N+1}}[y_N']\right). \tag{11}$$

## A.4 USING DMRG WITH $\text{MPS}_\sigma$

The combination of DMRG and $\text{MPS}_\sigma$ is not well defined. As shown in Figure 7, the last step of the DMRG algorithm involves performing SVD in order to obtain an optimal low-rank decomposition of the matricization of tensor $\widetilde{\boldsymbol{\mathcal{G}}}$, thereby solving

$$\arg\min_{\mathbf{G}^{(1)}, \mathbf{G}^{(2)}} \left\|\tilde{\mathbf{G}} - \mathbf{G}^{(1)}\mathbf{G}^{(2)}\right\|, \tag{12}$$

where $\mathbf{G}^{(1)} \in \mathbb{R}^{m \times r}$, $\mathbf{G}^{(2)} \in \mathbb{R}^{r \times n}$. However, in order for DMRG to apply to the $\mathrm{MPS}_\sigma$ class a different optimization problem must be solved, namely

$$\underset{\mathbf{G}^{(1)}, \mathbf{G}^{(2)}}{\arg \min} \left\| \tilde{\mathbf{G}} - \sigma\left(\mathbf{G}^{(1)}\right) \sigma\left(\mathbf{G}^{(2)}\right) \right\|. \tag{13}$$

## A.5 BACKWARD SAMPLING FROM AN MPS-BASED DISTRIBUTION

As MPS-based models have tractable marginals, inference of more sophisticated queries is possible. For example, backward auto-regressive sampling can be performed using

$$\begin{aligned} p(y_n|y_{n+1}, \ldots y_N) &= \frac{p(y_n, \ldots, y_N)}{p(y_{n+1}, \ldots, y_N)} \\ &= \frac{\dot{\boldsymbol{\mathcal{G}}}^{(1)}[y_1] \cdots \dot{\boldsymbol{\mathcal{G}}}^{(n-1)}[y_{n-1}] \boldsymbol{\mathcal{G}}^{(n)}[y_n] \boldsymbol{\mathcal{G}}^{(n+1)}[y_{n+1}] \cdots \boldsymbol{\mathcal{G}}^{(N)}[y_N]}{Z}. \end{aligned}$$

In Han et al. (2018), the authors provide examples of image inpainting by conditioning on particular subsets of inputs.

## A.6 COMPARISON BETWEEN LOG-SUM-EXP (LSE) AND LOGARITHMIC SCALE FACTORS (LSF)

While LSF bears resemblance to the Log-Sum-Exp (LSE) trick, there is a crucial difference. The Log-Sum-Exp trick cannot be used to train both Born Machines $\mathrm{MPS}_{\mathrm{BM}}$ or the large family of $\mathrm{MPS}_\sigma$ models for $\sigma \neq \exp$. However, *iterative* LSE and LSF are equivalent in the special case where $\sigma$ is chosen to be the exponential function.

Given the logarithm of a density function in the $\mathrm{MPS}_\sigma$ model class

$$\log p(y_1, \ldots, y_N) = \underbrace{\log\left(\sigma(\boldsymbol{\mathcal{G}}^{(1)})[y_1] \cdots \sigma(\boldsymbol{\mathcal{G}}^{(N)})[y_N]\right)}_{(A)} - \log Z \tag{14}$$

If $\sigma = \exp$ then we can use the **LSE**:

$$(A) = \log \sum \exp\left(\mathcal{G}^{(1)}_{r_1, r_2}[y_1] + \cdots + \mathcal{G}^{(N)}_{r_1, r_2}[y_N]\right) \tag{15}$$

$$= \log \sum \exp\left(\tilde{\mathcal{G}}^{(1)}_{r_1, r_2}[y_1] + \cdots + \tilde{\mathcal{G}}^{(N)}_{r_1, r_2}[y_N]\right) + \sum_i \alpha_i \tag{16}$$

where,

$$\alpha_i = \left\| \tilde{\mathcal{G}}^{(1)}[y_1] + \cdots + \tilde{\mathcal{G}}^{(i-1)}[y_{i-1}] + \mathcal{G}^{(i)}[y_i] \right\|_\infty \tag{17}$$

$$\tilde{\mathcal{G}}^{(i)} = \tilde{\mathcal{G}}^{(1)} + \cdots + \tilde{\mathcal{G}}^{(i-1)} + \mathcal{G}^{(i)} - \alpha_i \tag{18}$$

If $\sigma \neq \exp$ then we use **LSF**

$$(A) = \log \sum \left(\sigma(\mathcal{G}^{(1)}_{r_1, r_2}[y_1]) + \cdots + \sigma(\mathcal{G}^{(N)}_{r_1, r_2}[y_N])\right) \tag{19}$$

$$= \log \sum \left(\bar{\mathcal{G}}^{(1)}_{r_1, r_2}[y_1] + \cdots + \bar{\mathcal{G}}^{(N)}_{r_1, r_2}[y_N]\right) + \sum_i \log \gamma_i \tag{20}$$

where

$$\gamma_i = \left\| \bar{\mathcal{G}}^{(1)}[y_1] \cdots \bar{\mathcal{G}}^{(i-1)}[y_{i-1}] \sigma(\mathcal{G}^{(i)}[y_i]) \right\|_\infty \tag{21}$$

$$\bar{\mathcal{G}}^{(i)} = \frac{1}{\gamma_i} \bar{\mathcal{G}}^{(1)} \cdots \bar{\mathcal{G}}^{(i-1)} \sigma(\mathcal{G}^{(i)}) \tag{22}$$

Figure 7: (Reproduction of Figure 4) Illustration of a single update step using the DMRG two site update algorithm used in Han et al. (2018). (1) cores $\boldsymbol{\mathcal{G}}^{(1)}$ and $\boldsymbol{\mathcal{G}}^{(2)}$ are merged (2) the loss is computed with respect to the merged fourth order tensor (3) the gradient is computed and used to update the fourth order tensor using automatic differentiation (4) the fourth order tensor is decomposed using SVD, then singular vectors are *copied* into cores $\boldsymbol{\mathcal{G}}^{(1)}$ and $\boldsymbol{\mathcal{G}}^{(2)}$.

## A.7 PROOFS

**Lemma 1.** *Let $X$ denote a normally distributed random variable, $\sigma : \mathbb{R} \to \mathbb{R}_{\geq 0}$ denote a non-negative mapping s.t.*

$$\forall x \in \mathbb{R}_{>0}, \; \exists \, \epsilon_x > 0 \;\; s.t. \;\; \sigma(x) > \epsilon_x.$$

*Then, $\exists \, \epsilon > 0$ s.t. $\mathbb{E}_X \left[ \sigma(x) \right] > \epsilon$*

*Proof.* Let $a, b \in \mathbb{R}$ and $0 < a < b$. Then,

$$\mathbb{E}_X \left[ \sigma(x) \right] = \int \sigma(x) f_X(x) \tag{23}$$

$$\geq \int_a^b \sigma(x) f(x) \tag{24}$$

$$\geq \inf\{\sigma(x) \,|\, a \leq x \leq b\} \int_a^b f_X(x) \tag{25}$$

$$= \epsilon' \tag{26}$$

$$\geq \frac{\epsilon'}{2} \tag{27}$$

$$= \epsilon \tag{28}$$

where Equation 24 holds because both $\sigma$ and $f_X$ are non-negative, Equation 26 holds because both $\sigma(x) > 0$ and $f_X(x) > 0$ for $x \in [a, b]$. Lastly, we have that $\epsilon > 0$ since $\epsilon' > 0$. $\qquad \square$

**Theorem 1.** *Let the elements of the tensor $\boldsymbol{\mathcal{G}}^{(i)} \in \mathbb{R}^{R_i \times D \times R_{i+1}}$ be i.i.d. random variables drawn from a zero-mean gaussian distribution with unit variance, $R_1 = R_N = 1$ and $R_i = R \quad \forall i \neq 1, N$. Let $\mathbf{y} \in \mathcal{Y}$, $\mathcal{Y} = \mathcal{Y}_i \times \cdots \times \mathcal{Y}_N$ and*

$$\Psi_\sigma(\mathbf{y}) = \sigma(\boldsymbol{\mathcal{G}}^{(1)}[y_1]) \cdots \sigma(\boldsymbol{\mathcal{G}}^{(N)}[y_N]), \quad Z_\sigma = \dot{\boldsymbol{\mathcal{G}}}^{(1)} \cdots \dot{\boldsymbol{\mathcal{G}}}^{(N)}, \tag{6}$$

*where $\dot{\boldsymbol{\mathcal{G}}}_{ij} \triangleq \sum_k \sigma(\boldsymbol{\mathcal{G}}_{ikj})$ and $\sigma : \mathbb{R} \to \mathbb{R}_{\geq 0}$ is a point-wise non-negative mapping s.t.*

$$\forall x \in \mathbb{R}_{>0}, \; \exists \, \epsilon_x > 0 \;\; s.t. \;\; \sigma(x) > \epsilon_x.$$

*Then, $\mathbb{E}[\Psi_\sigma(\mathbf{y})] \geq \epsilon R^N$ and $\mathbb{E}[Z_\sigma] \geq \epsilon R^N D^N$ for some $\epsilon > 0$*

*Proof.* Let $\mathcal{R} = \{n \mid n \in \mathbb{N}, n \leq R\}$ denote a set of integers, then the expected value of $\Psi_\sigma(\mathbf{y})$ is bounded below, since

$$\mathbb{E}[\Psi_\sigma(\mathbf{y})] = \mathbb{E}\left[\sum_{\mathbf{r} \in \mathcal{R}^N} \sigma\left(\boldsymbol{\mathcal{G}}^{(1)}_{r_1 r_2}[y_1]\right) \cdots \sigma\left(\boldsymbol{\mathcal{G}}^{(N)}_{r_N r_{N+1}}[y_N]\right)\right] \tag{29}$$

$$= \sum_{\mathbf{r} \in \mathcal{R}^N} \mathbb{E}\left[\sigma\left(\boldsymbol{\mathcal{G}}^{(1)}_{r_1 r_2}[y_1]\right)\right] \cdots \mathbb{E}\left[\sigma\left(\boldsymbol{\mathcal{G}}^{(N)}_{r_N r_{N+1}}[y_N]\right)\right] \tag{30}$$

$$> \sum_{\mathbf{r} \in \mathcal{R}^N} \epsilon^{(r_1)} \cdots \epsilon^{(r_N)} \tag{31}$$

$$\geq \sum_{\mathbf{r} \in \mathcal{R}^N} \tilde{\epsilon}^N \tag{32}$$

$$= \epsilon R^N, \tag{33}$$

where $\tilde{\epsilon} \triangleq \min \epsilon^{(r_1)} \cdots \epsilon^{(r_N)}$ and Equation 31 follows from Lemma 1a . Similarly, the normalization constant is bounded below,

$$\mathbb{E}[Z_\sigma] = \mathbb{E}\left[\sum_{\mathbf{r} \in \mathcal{R}^N} \sum_{\mathbf{y} \in \mathcal{Y}} \sigma\left(\boldsymbol{\mathcal{G}}^{(1)}_{r_1 r_2}[y_1]\right) \cdots \sigma\left(\boldsymbol{\mathcal{G}}^{(N)}_{r_N r_{N+1}}[y_N]\right)\right] \tag{34}$$

$$= \sum_{\mathbf{r} \in \mathcal{R}^N} \sum_{\mathbf{y} \in \mathcal{Y}} \mathbb{E}\left[\sigma\left(\boldsymbol{\mathcal{G}}^{(1)}_{r_1 r_2}[y_1]\right)\right] \cdots \mathbb{E}\left[\sigma\left(\boldsymbol{\mathcal{G}}^{(N)}_{r_N r_{N+1}}[y_N]\right)\right] \tag{35}$$

$$> \sum_{\mathbf{r} \in \mathcal{R}^N} \sum_{\mathbf{y} \in \mathcal{Y}} \epsilon^{(1)} \cdots \epsilon^{(N)} \tag{36}$$

$$\geq \sum_{\mathbf{r} \in \mathcal{R}^N} \sum_{\mathbf{y} \in \mathcal{Y}} \tilde{\epsilon}^N \tag{37}$$

$$= \epsilon R^N D^N. \tag{38}$$

$\square$

**Theorem 2.** *Let the elements of the tensor $\boldsymbol{\mathcal{G}}^{(i)} \in \mathbb{R}^{R_i \times D \times R_{i+1}}$ be i.i.d. random variables drawn from a zero-mean gaussian distribution with unit variance, $R_1 = R_N = 1$ and $R_i = R \quad \forall i \neq 1, N$. Let $\mathbf{y} \in \mathcal{Y}$, $\mathcal{Y} = \mathcal{Y}_i \times \cdots \times \mathcal{Y}_N$ and $\Psi_{\mathrm{BM}}(\mathbf{y}) = \boldsymbol{\mathcal{G}}^{(1)}[y_1] \cdots \boldsymbol{\mathcal{G}}^{(N)}[y_N]$. Then, $\mathbb{E}[\Psi_{\mathrm{BM}}(\mathbf{y})] = 0$ and $\mathbb{E}[\Psi_{\mathrm{BM}}(\mathbf{y})^2] \geq \epsilon R^N$ for some $\epsilon > 0$*

*Proof.* We have that the expectation of $\Psi_{\mathrm{BM}}(\mathbf{y})$ is zero since,

$$\mathbb{E}[\Psi_{\mathrm{BM}}(\mathbf{y})] = \mathbb{E}\left[\sum_{\mathbf{r} \in \mathcal{R}^N} \boldsymbol{\mathcal{G}}^{(1)}_{r_1 r_2}[y_1] \cdots \boldsymbol{\mathcal{G}}^{(N)}_{r_N r_{N+1}}[y_N]\right] \tag{39}$$

$$= \sum_{\mathbf{r} \in \mathcal{R}^N} \mathbb{E}\left[\boldsymbol{\mathcal{G}}^{(1)}_{r_1 r_2}[y_1]\right] \cdots \mathbb{E}\left[\sigma\left(\boldsymbol{\mathcal{G}}^{(N)}_{r_N r_{N+1}}[y_N]\right)\right] \tag{40}$$

$$= 0. \tag{41}$$

Therefore, its variance is given by

$$\mathbb{E}[\Psi_{\text{BM}}(\mathbf{y})^2] = \int_{\boldsymbol{\mathcal{G}}} \left( \sum_{\mathbf{r}\in\mathcal{R}} \boldsymbol{\mathcal{G}}^{(1)}_{r_1 r_2}[y_1]\cdots\boldsymbol{\mathcal{G}}^{(N)}_{r_N r_{N+1}}[y_1] \right)^2 f_{\mathcal{G}}(\boldsymbol{\mathcal{G}})\,d\boldsymbol{\mathcal{G}} \tag{42}$$

$$= \frac{C}{2} + \int_{\boldsymbol{\mathcal{G}}\in\mathcal{S}} \left( \sum_{\mathbf{r}\in\mathcal{R}} \boldsymbol{\mathcal{G}}^{(1)}_{r_1 r_2}[y_1]\cdots\boldsymbol{\mathcal{G}}^{(N)}_{r_N r_{N+1}}[y_1] \right)^2 f_{\mathcal{G}}(\boldsymbol{\mathcal{G}})\,d\boldsymbol{\mathcal{G}} \tag{43}$$

$$\geq \int_{\boldsymbol{\mathcal{G}}\in\mathcal{S}} \sum_{\mathbf{r}\in\mathcal{R}} \boldsymbol{\mathcal{G}}^{(1)}_{r_1 r_2}[y_1]^2\cdots\boldsymbol{\mathcal{G}}^{(N)}_{r_1 r_2}[y_1]^2 f_{\mathcal{G}}(\boldsymbol{\mathcal{G}})f_{\mathcal{G}}(\boldsymbol{\mathcal{G}})\,d\boldsymbol{\mathcal{G}} \tag{44}$$

$$= \int_{\boldsymbol{\mathcal{G}}\in\mathcal{S}} \sum_{\mathbf{r}\in\mathcal{R}} \sigma\left(\boldsymbol{\mathcal{G}}^{(1)}_{r_1 2}[y_1]\right)\cdots\sigma\left(\boldsymbol{\mathcal{G}}^{(N)}_{r_1 2}[y_1]\right) f_{\mathcal{G}}(\boldsymbol{\mathcal{G}})\,d\boldsymbol{\mathcal{G}} \tag{45}$$

$$= \int_{\boldsymbol{\mathcal{G}}\in\mathcal{S}} \sum_{\mathbf{r}\in\mathcal{R}} \epsilon^{(r_1,r_2)}\cdots\epsilon^{(r_N,r_{N+1})} f_{\mathcal{G}}(\boldsymbol{\mathcal{G}})\,d\boldsymbol{\mathcal{G}} \tag{46}$$

$$= \frac{1}{2}\sum_{\mathbf{r}\in\mathcal{R}} \tilde{\epsilon} \tag{47}$$

$$= \frac{1}{2}\tilde{\epsilon}R^H \tag{48}$$

$$= \epsilon R^H, \tag{49}$$

where $\mathcal{S}$ represents the infinite set consisting of all parameters $\boldsymbol{\mathcal{G}} \triangleq \{\boldsymbol{\mathcal{G}}^{(N)}\ldots\boldsymbol{\mathcal{G}}^{(N)}\}$ that result in a positive contraction at test point $\mathbf{y}$. Thus, equation 43 follows by symmetry of the distribution represented by a product of $N$ zero-mean independent gaussian random variables. $\qquad\square$

## A.8 QUALITATIVE RESULTS

In Figure 8, we plot samples from the $\text{MPS}_\sigma$ model at various checkpoints. For all checkpoints overfitting was not observed (i.e. validation loss was still monotonically decreasing). While the samples plotted in Figure 8 are not comparable to state of the art image generation models, we highlight that qualitatively they are comparable to samples generated with other MPS-based models trained using DMRG (Han et al., 2018).

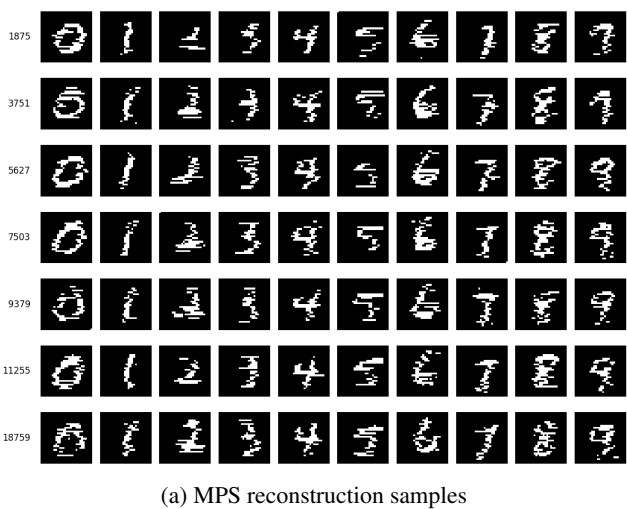

(a) MPS reconstruction samples

(b) Ground truth MNIST samples

Figure 8: Comparison between MPS-generated MNIST samples and ground truth MNIST images.

