# OpenReview forum: "Efficient Learning of Probabilistic Tensor Networks"
_ICLR.cc/2026/Conference — Submitted to ICLR 2026_

### Official Review · Reviewer_v1ua · 2025-10-17

**Soundness:** 2
**Presentation:** 3
**Contribution:** 2
**Rating:** 4
**Confidence:** 4

**Summary:**

This work studies the problem of gradient-based training of probabilistic tensor networks, e.g., MPS. Directly using gradient-based optimizers would be very unstable for long MPS, due to the long sequence of matrix multiplication. This paper proposes a method, which normalizes every matrix factor, and then adds the scaling factor on the logarithm. Empirically, the authors show that this scaling makes the training stable, enabling gradient-based training of long MPS, which is impossible for SGD.

**Strengths:**

1. I think the problem of gradient-based optimization of MPS is meaningful and important in this field. As the authors introduced, the unstable training using SGD and the complexity in both training and implementation of DMRG have been a bottleneck of applying such methods.

2. The proposed method is simple to use, does not introduce additional latency, and seems to be much more stable than SGD.

**Weaknesses:**

1. My main concern is that a similar trick has been applied to existing libraries. The ContractTN (https://github.com/jemisjoky/ContracTN/blob/main/ContracTN_QTNML_2021_Workshop.pdf) library also uses the rescaling in the logarithm for numerical stability. The scaling factor is also based on the norm of the tensor factors. Could the authors compare with it and emphasize the difference?

2. The analysis in Section 3.4 seems nonrigorous. The authors assume the elements in the tensor factors follow standard Gaussian distribution with unit variance. This assumption needs to be justified. When training neural networks, we need to carefully adjust the initialization and learning rates for a stable training, e.g., https://arxiv.org/pdf/2310.17813. I guess it is also true for training MPS.

3. The authors do not mention the details of the baseline SGD. Using the TensorNetwork library (https://github.com/google/TensorNetwork), it can also achieve competitive performance on high-dimensional data like MNIST simply using Adam (https://arxiv.org/pdf/1906.06329). So I think more comparisons with existing libraries would be useful.

4. When training neural networks, normalization techniques like layer and batch normalization are important. So I am curious how would simply applying these techniques compare with the proposed one.

**Questions:**

Please see above

---

> ### Author Response · Authors · 2025-11-21
> **Response to weaknesses (novelty, gaussian assumption, baseline details, batch norm)**
>
> We would like to thank the reviewer for their feedback and agree with the reviewer regarding gradient-based optimization of MPS being meaningful and important in this field. Please see below our responses to the points you have raised:
>
> ***W1: My main concern is that a similar trick has been applied to existing libraries. The ContractTN***
>
> Thank you for bringing the work to our attention. We have reviewed this workshop paper carefully and indeed the trick presented in ContractTN is the same as ours. We have cited this workshop paper and highlighted that our work builds on top of this trick. While the trick is the same, the workshop paper did not theoretically analyze the numerical instability that arises in PTNs and did not provide experimental results to support the claim of that this trick sufficiently addresses instability issues in real-world tasks. Specifically, we have revised our contributions to be:
>
> 1. **Theoretical analysis:** We provide a formal characterization of the numerical instabilities that arise when training PTNs such as $\mathrm{MPS}\_\sigma$ and $\mathrm{MPS}\_{\mathrm{BM}}$. In particular, we prove lower bounds on the mean and variance of the computations associated with members of both classes, respectively.
> 2. **Experimental results:** We demonstrate empirically that, on real datasets, our stabilized training procedure achieves performance competitive with DMRG (a significantly more stable but computationally heavier algorithm than SGD) while running approximately **10× faster**.
>
> ***W2: The analysis in Section 3.4 seems nonrigorous. The authors assume the elements in the tensor factors follow standard Gaussian distribution with unit variance.***
>
> Indeed while the normal distribution assumption does not remain valid throughout training, the purpose of this theorem is to characterize the behaviour of the model at initialization. In particular, the theorem establishes that already during the first forward pass a numerical overflow occurs unless an appropriate stabilization mechanism is employed.
>
> ***W3: The authors do not mention the details of the baseline SGD. Using the TensorNetwork library can also achieve competitive performance on high-dimensional data:***
>
> We have updated our appendix to include the experimental details A.1.2, A.1.3 as well as updated Section 4.3. Specifically for the density estimation benchmarks we used Adam in place of SGD as it yielded better performance.
>
> Regarding the TensorNetwork library, the link you shared does not seem to contain an example train script used for training on MNIST. We would like to highlight that as this work does not make reference to additional stabilization strategies that the authors may have resorted to using higher numerical precision (FP64) while using vanilla SGD (or Adam) for training an MPS network on MNIST sized problems.
>
> ***W4: When training neural networks, normalization techniques like layer and batch normalization are important. So I am curious how would simply applying these techniques compare with the proposed one.***
>
> This is an interesting question. Applying batch norm in the context of density estimation with MPS does not appear to be straightforward since batch norm involves both subtraction by the batch mean as well as division by the batch std deviation. Consider the case of $\mathrm{MPS}_\sigma$ (for brevity we assume all cores are positive). Then:
>
> $$\log p(\mathbf x) = \log G^{(1)}\_{x_1} \cdots G^{(N)}\_{x_N} - \log Z$$
>
> Now applying batch norm at the $k^\text{th}$ step would amount to standardizing an R-dimensional vector $\widetilde{G}^{(1)}\_{x\_1}\cdots
> \widetilde{G}^{(k-1)}\_{x\_{k-1}}
> G^{(k)}\_{x\_{k}}$. While the division by the standard deviation can be performed and extracted in a similar manner as LSF, subtraction by the mean is tricker. The same subtraction by mean performed for the unnormalized probability must be **identically applied** to the normalization constant $Z$. While in principle this can work, this adds dependence between the computation of the unnormalized probability and the computation of the normalization constant, **making the overall computation less parallelizable.**

---

### Official Review · Reviewer_DvCY · 2025-10-30

**Soundness:** 3
**Presentation:** 3
**Contribution:** 2
**Rating:** 2
**Confidence:** 4

**Summary:**

The paper consider density estimation by non-negative tensor network representations based on stochastic gradient descent optimization stabilized through logarithmic scaling factors enabling numerically stable and efficient learning of very large densities. The approach is compared to DMRG for Borne Machines (i.e. squared representations to form non-negative tensors) as well as previous work using SGD for optimizing squared representations as well as imposing a non-negative tensor network factorization demonstrating that the existing SGD based approach fail whereas the proposed LSF works when having many cores being computationally efficient when compared to DMRG based inference.

**Strengths:**

The approach enables simple SGD learning of non-negative tensor network decompositions for density estimation ensuring numerical stability and enabling large scale density estimation which is useful. The approach is further very simple to implement.

Originality: The approach is in my opinion not original and essentially the logarithmic scaling factors is very similar to the widely used logsumexp operation to achieve numerical stability when evaluating the log of densities constituted by non-negative sums (typically exponentiated but here formed by applying the \sigma operator to project from unconstrained to positive elements in the non-negative decomposition) with the log scaling factor similarly ensuring numerical stability.

Quality: The paper is insufficiently related to existing modeling procedures for density estimation by use of tensor decompositions and should include more extensive comparisons.

Clarity: The paper is clearly written and generally easy to follow.

Significance: The results are compelling but the novelty is very limited of the procedure and straightforward. Furthermore, existing non-negative tensor decomposition procedures relying on the probabilistic circuit formulation does not suffer as far as I understand of the described problems of inability to scale to large dimensions and being hampered by numerical instability. I also believe the existing tensor train density estimation TTDE framework (see below) as part of their code address numerical instabilities. I thus find the papers contribution very limited and not sufficiently positioned in the literature.

**Weaknesses:**

The main weakness is that the approach is straightforward and the contribution of limited novelty. Although the approach is practical useful it is very simple and straightforward and using what appears to be very close to existing standard procedures to achieve numerical stability of density estimates exploring their log-domain representations and how log-scaling factors through the decomposition can decouple into factors suitable for numerical stable inference.

**Questions:**

There is a vast literature on tensor network based density estimation that is not accounted for and that can operate on high-dimensional problems to my understanding without numerical issues. The paper needs to carefully position and establish this work also in relation to this literature, that includes:
TTDE: Tensor Train Density Estimation using Riemmannian optimization considering the least squares objective and squared representation for non-negative tensors as presently considered for tensor train density estimation (TTD) i.e. identical to the presented MPS BM formalism in model structure.
https://proceedings.mlr.press/v161/novikov21a/novikov21a.pdf
This procedure to my understanding use similar tricks to stabilize inference by suitable scaling factors. Please check their code and compare how their procedure performs to yours in the high-dimensinal setting.

Tensor Ring Density Estimation (TRDE/TERM): The tensor ring density estimation procedure described n in:
https://arxiv.org/abs/2312.08075
considers log-likelihood as opposed to least squares loss minimization and should extend to high-dimensional problems.

Tensor network based probabilistic circuits: There is a recent review of probabilistic circuit and its relation to tensor network representations for density estimation including code examples etc., see:

Loconte, Lorenzo, et al. "What is the Relationship between Tensor Factorizations and Circuits (and How Can We Exploit it)?." Transactions on Machine Learning Research (02/2025).

Notably such non-negative decompositions using probabilistic circuits also rely on non-negative tensor network representations but by additionally inducing distributional constraints on the carts this ensures that the resulting tensor-network representation naturally form valid densities. In this context it is also unclear why similar constraints on the non-negative carts in equation 5 cannot be imposed? Such constraints also naturally handle the numerical instabilities encountered when fitting models and makes learning probabilistic circuit based tensor network structures even when the densities are very high-dimensional possible without numeric issues. In fact the log-scaling factors used in LSF seem to indirectly also explore this property. Tensor factorization based probabilitic circuits including the tensor train formalism has also been considered in:

N. Ciolli, M. Mørup and M. N. Schmidt, "TNSPC: Learning from Partially Observed Data Using Tensor Network Structured Probabilistic Circuits," 2025 IEEE 35th International Workshop on Machine Learning for Signal Processing (MLSP), doi: 10.1109/MLSP62443.2025.11204307.

with code provided considering also MNIST sized problems in the GitHub repository.

Non-negative tensor factorization procedures for density estimation: Non-negative tensor factorizations have been widely used for density estimation with associated EM-based procedures used for stable inference,  see also the framework presented in

https://arxiv.org/pdf/2405.18220

and references therein. These methods can also exploit that the non-negative decompositions naturally can be expressed by suitably normalized carts which also ensures numerical stability and scaling to high-dimensional problem in my understanding.

---

> ### Author Response · Authors · 2025-11-21
> **Response to weaknesses and questions (novelty and position with respect to the literature)**
>
> We would like to thank the reviewer for their thorough feedback. Please see below our responses to the points you have raised:
>
> ***W1: The main weakness is that the approach is straightforward and the contribution of limited novelty.***
>
> We appreciate the reviewer’s concern and agree that the LSF mechanism itself is straightforward. We have revised our contribution statement in the introduction to more clearly articulate the specific claims we make. Our contributions are:
>
> 1. **Theoretical analysis:** We provide a formal characterization of the numerical instabilities that arise when training PTNs such as $\mathrm{MPS}\_\sigma$ and $\mathrm{MPS}\_{\mathrm{BM}}$. In particular, we prove lower bounds on the mean and variance of the computations associated with members of both classes, respectively.
> 2. **Experimental results:** We demonstrate empirically that, on real datasets, our stabilized training procedure achieves performance competitive with DMRG (a significantly more stable but computationally heavier algorithm than SGD) while running approximately **10× faster**.
>
> We have updated our introduction to reflect these changes.
>
> ***Q1: There is a vast literature on tensor network based density estimation that is not accounted for and that can operate on high-dimensional problems to my understanding without numerical issues.***
>
> We have cited the above works and described the differences between our work and the ones cited in the updated PDF. The changes were made in the related work section. We would like to highlight that the cited works do not discuss numerical instability issues and in many cases have experiments limited to a small number of variables. We discuss in our paper that in practice, numerical instabilities arise only on problems with over 100 random variables. We summarize the differences with the cited papers here:
>
> **[1] TTDE ([Novikov et al., 2021](https://arxiv.org/abs/2108.00089))**
>
> - Real datasets: 5 (POWER, GAS, HEPMASS, MINIBOONE, BSDS300)
> - Max dim of real data: **64 features**
>
> **[2] TERM ([Wu et al., 2023](https://arxiv.org/pdf/2312.08075))**
>
> - Real datasets: 4 (POWER, GAS, HEPMASS, MINIBOONE)
> - Max dim of real data: **9 features**
>
> **[3] E2M (**[Ghalamkari et al., 2024](https://arxiv.org/pdf/2405.18220)**)**
>
> - Real datasets: 34
> - Max dim of real data (Table 5): **36 features**
>
> **[4] Tensors & PCs et. al ([Laconte et al., 2024](https://arxiv.org/abs/2409.07953))**
>
> - This paper indeed has MNIST sized problems. However, this work implements their methods as probabilistic circuits and does not demonstrate a numerical stabilization technique for Born Machines.
>
> **[5] TNSPC ([Cioli et al., 2025](https://ieeexplore.ieee.org/document/11204307))**
>
> - This paper indeed has larger problem sizes (1552 variables on the AD dataset). However, this work also implements their methods as probabilistic circuits and does not provide a numerical stabilization technique for Born Machines.
>
>
> ***Q2: Is this just a direct application of Log-Sum-Exp trick?***
>
> We would like to clarify that the stabilization procedure in Section 3.5 is *not* simply an application of the Log-Sum-Exp (LSE) trick. In particular, the LSE trick **cannot be directly employed for Born Machines**, since their probabilities are defined via squared amplitudes rather than through enforcing positivity by exponentiating individual MPS cores.
>
> In addition to Born Machines, our stabilization procedure applies to any positive MPS parameterization (i.e., the $\mathrm{MPS}\_\sigma$ class), including those employing non-exponential activations (e.g., Absolute, Square, or Sigmoid). For these choices of $\sigma$, the classical Log-Sum-Exp (LSE) trick is not directly applicable. Only in the special case $\sigma=\exp$ does our method reduce to the standard LSE-based stabilization, which also explains its resemblance to techniques used in works such as EiNets that use the exponential function for positivity enforcement.
>
> Lastly, we have revised Section 3.5 to better illustrate the generality of the approach, by demonstrating its applicability to the Born Machine as well as added a paragraph clarifying the conditions under which LSF and LSE are equivalent and expanded on it in Appendix A.5

---

> > ### Comment · Reviewer_DvCY · 2025-11-24
> >
> > My concern regarding this paper remains in light of the authors' rebuttals.
> > I find the contribution still to be straightforward, of limited novelty and not substantial enough to recommend publication.
> > The authors in their rebuttal also did not address my concern regarding TTDE and did not investigate if the code here provided already in the TTDE implementation also address the numerical stability issues presently discussed. I agree the applications in TTDE are on lower dimensional problems but I believe their code can easily be applied to high-dimensional problems and the main reason was comparison to some standard benchmarks for density estimation.
> > The authors need to discuss how their approach compare to this existing work with code available. Also as pointed out by reviewer v1ua and acknowledged by the authors the approach to stabilize training is not even new and covered in ContractTN with the paper presently providing theoretical analysis and extensive experimental evaluations.
> > I consequently still find the contribution straightforward, of limited novelty, and not substantial enough to recommend publication. The theoretical analysis and extensive experimentation is a fine contribution but not in my view warranting to recommend this paper for publication at ICLR.

---

> > > ### Author Response · Authors · 2025-11-30
> > > **Response to not addressing TTDE and ContractTN**
> > >
> > > We thank the reviewer for their comment. We believe we have addressed the reviewer's comment regarding TTDE. As shown in our work, numerical issues only arise over **100+** variables meanwhile TTDE was limited to only **64** variables. In TTDE the authors do not discuss any numerical issues that arise when training using SGD/Adam, likely because their experimental setting consisted of only 64 variables and did not lead them to encounter such numerical issues. Moreover, we did not see any additional steps taken by the authors in their codebase to mitigate numerical issues that would arise at large variable sizes when training with SGD/Adam.
> > >
> > > Regarding ContractTN, we would like to highlight that ContractTN was a workshop paper that did not provide any theoretical results. Furthermore, ContractTN did not provide any experimental results. In contrast, we derive lower bounds that characterize the numerical instabilities in PTNs and demonstrate that LSF can effectively mitigate these issues on datasets with large variable sizes (**1000+**)

---

### Official Review · Reviewer_9gj5 · 2025-10-31

**Soundness:** 2
**Presentation:** 3
**Contribution:** 2
**Rating:** 4
**Confidence:** 3

**Summary:**

This paper proposes a numerically stable and fully automatic-differentiation-compatible method for training probabilistic tensor networks (PTNs) based on Matrix Product States (MPS). The authors identify that prior approaches—DMRG and vanilla SGD—either incur high computational cost or suffer from numerical instability. They introduce a logarithmic-scale-factor (LSF) training scheme that stabilizes gradient computation, enabling efficient end-to-end learning within mainstream deep-learning frameworks. Experiments demonstrate significant speed and memory gains, along with the ability to train on sequences 10× longer than before, while achieving competitive likelihood performance on benchmarks including MNIST and density estimation datasets.

**Strengths:**

1. Well-written paper which addresses critical limitations of prior PTN training methods, namely numerical instability and incompatibility with automatic differentiation (e.g., DMRG overhead) .

2. Analyzes SGD instability for PTNs and provides bounds showing exponential scaling in instability, motivating the method.

**Weaknesses:**

1. My major concern is about the application breadth: While MNIST and tabular benchmarks are reasonable, evaluation on larger or more modern datasets (e.g., CIFAR, text) would strengthen claims of broad usability. Could the author provide more experimental results on datasets such as CIFAR and ImageNet?

2. The MNIST experiment reports negative log-likelihood but does not include quality metrics such as FID. Given that the task is described as generative modeling, including FID (or qualitative samples) would strengthen the evaluation and demonstrate visual fidelity, not just likelihood performance.

3. In Table 3, not all MPS+LSF variants achieve large speedups. Can the authors explain when LSF is most critical and when its advantage shrinks?

4. (Minor) In line 457-458, it is said MPS_{\sigma+LSF} achieves 10x speedup than MPS_{BM+DMRG}. But in Table 3, it is MPS_{BM+LSF} has 10x speed up than MPS_{BM+DMRG} right?

5. Can the LSF scheme be applied to other tensor-network architectures (e.g., TT, TR, PEPS, MERA)? Or is it specific to the chain structure of MPS?

**Questions:**

Please see the weakness above.

---

> ### Author Response · Authors · 2025-11-21
> **Response to weaknesses (breadth of experiments, qualitative evaluation, clarification on speedup)**
>
> We would like to thank the reviewer for their feedback and their positive remarks regarding the clarity of our writing in addition to them highlighting that our method addresses the exponential scaling in instability that occurs during the training of PTNs.
>
> ***W1: My major concern is about the application breadth … could the author provide more experimental results on datasets such as CIFAR and ImageNet?***
>
> We have updated Table 4 and added a direct comparison between our Log-Scale Factors (LSF) method and DMRG across the **20 density estimation benchmarks**.
>
> We would like to emphasize that our experimental scope follows the conventional setting in tensor-based density estimation, which predominantly considers binary-valued variables (see references). However, we note that prior work using SGD-based training has evaluated such models only on real datasets containing **fewer than 65 variables**. In contrast, we show that our approach remains effective on real datasets with **well over 1000 variables**, demonstrating a substantial increase in scale.
>
> - *MPS expressiveness* ([Glasser et al., 2019](https://arxiv.org/abs/1907.03741)): **22** variables
> - *TTDE* ([Novikov et al., 2021](https://arxiv.org/abs/2108.00089)): **64** variables
> - *TERM* ([Wu et al., 2023](https://arxiv.org/pdf/2312.08075)): **9** variables
> - *Born Machines* ([Han et al., 2019](https://arxiv.org/abs/1709.01662)) — trained with **DMRG**
> - *Tensor Tree Networks* ([Liu et al., 2019](https://arxiv.org/abs/1901.02217)) — trained with **DMRG**
>
> ***W2 The MNIST experiment reports negative log-likelihood but does not include quality metrics … including FID (or qualitative samples) would strengthen the evaluation***
>
> We have made an update to the PDF file to include qualitative samples (Figure 8). We would like to highlight that qualitatively LSF yields similar results to those obtained using DMRG in ([Han et al., 2019](https://arxiv.org/abs/1709.01662))
>
> ***W3: In Table 3, not all MPS+LSF variants achieve large speedups. Can the authors explain when LSF is most critical and when its advantage shrinks?***
>
> We made a typo in this table for one of the results (0.98) ⇒ (0.12), all the MPS+LSF variants have comparable latency
>
> ***W4: (Minor) In line 457-458, it is said $\mathrm{MPS}\_{\sigma+\mathrm{LSF}}$ achieves 10x speedup than $\mathrm{MPS}\_\mathrm{BM+DMRG}$. But in Table 3, it is $\mathrm{MPS}\_\mathrm{BM+LSF}$ has 10x speed up than $\mathrm{MPS}\_\mathrm{BM+DMRG}$ right?***
>
> Following from the previous question, we made a typo in this table for one of the results that is now fixed. All the $\mathrm{MPS}\_{\sigma+\mathrm{LSF}}$ variants as well as the $\mathrm{MPS}\_\mathrm{BM+LSF}$  have comparable latency
>
> ***W5: Can the LSF scheme be applied to other tensor-network architectures (e.g., TT, TR, PEPS, MERA)? Or is it specific to the chain structure of MPS?***
>
> Indeed it can be applied, while we do not handle the treatment of each TN structure we updated the conclusion section to highlight this as an area of future work.

---

### Official Review · Reviewer_R7nx · 2025-11-09

**Soundness:** 3
**Presentation:** 3
**Contribution:** 2
**Rating:** 4
**Confidence:** 4

**Summary:**

In this paper the authors theoretically analyze the cause of numerical stability when learning parameters of MPS-based TPNs used SGD. Consequently, they develop a numerically stable approach for computing the negative log-likelihood through the use of logarithmic scale factors. They then show that their developed approach cab be used to process sequences that are 10x longer than managed by previous SGD-based approaches, while being 10x faster than alternative, numerically stable approaches relying on DMRG.

**Strengths:**

- The paper is overall well-written

- The authors clearly motivate the problem they're solving, supporting their motivation for the problem with empirical insights.

- The proposed approach avoids numerical overflow suffered by using SGD.

- The proposed approach leads to a 10x drop in latency compare to DMRG at a negligible decrease in negative log-likelihood

**Weaknesses:**

- I think the authors would benefit from summarizing their approach in the abstract. As it stand, the only information a reader is able to extract from the abstract is that the authors propose a simple approach for efficiently learning probabilistic tensor networks.

- In my opinion, the authors provide very little by way of intuition and hand-holding in section 3.5, which I understand to be their main contribution. A toy example with a figure would've been very helpful here as well. Moreover, I would've expected the authors to highlight the properties of the probabilistic models that make this possible, since I presume that these local normalizations are only possible due to the form of the MPS model.

- Consequently, I am left feeling a bit skeptical that the contribution in section 3.5 would be novel, as it is my understanding that it is simply a straight-forward application of the logsumexp trick, although I am happy to be corrected. For example, I believe that the EiNets paper makes use of a similar trick.

- I would've like to see more of a comparison between the proposed approach and DMRF in terms of NLL, in addition to MNIST.

**Questions:**

- What does line 4 in Figure 1 (a) correspond to? Does that denote matrix multiplication?

- The authors mention that DMRG is non-differentiable. I expected that to translate to some form or biased relaxation, and therefore a higher NLL. However, that does not seem to be the case, at least on the MNIST experiments.

- Please see the "Weaknesses" section for a list of other questions/concerns.

---

> ### Author Response · Authors · 2025-11-21
> **Response to weaknesses (improving method section, addressing novelty, increasing experiments)**
>
> We would like to thank the reviewer for their feedback and their positive remarks regarding the clarity of our writing, as well as pointing out that our method provides a 10x drop in latency compared to DMRG. We provide responses to each of the weaknesses and the questions the reviewer has raised below:
>
>
> ***W1: I think the authors would benefit from summarizing their approach in the abstract***
>
> We have modified the abstract to include more information on the proposed method. Thank you for your feedback.
>
> ***W2: In my opinion, the authors provide very little by way of intuition and hand-holding in section 3.5, which I understand to be their main contribution***
>
> We have **revised Section 3.5** to better introduce our method. We first show that products of scale factors can be added without changing the loss function, then describe our particular choice. We have also **updated Figure 3** to provide evidence for on the efficacy of this choice of scale factors earlier on in the paper.
>
> ***W3: Consequently, I am left feeling a bit skeptical that the contribution in section 3.5 would be novel, as it is my understanding that it is simply a straight-forward application of the logsumexp trick…***
>
> We would like to clarify that the stabilization procedure in Section 3.5 is *not* simply an application of the Log-Sum-Exp (LSE) trick. In particular, the LSE trick **cannot be directly employed for Born Machines**, since their probabilities are defined via squared amplitudes rather than through enforcing positivity by exponentiating individual MPS cores.
>
> In addition to Born Machines, our stabilization procedure applies to any positive MPS parameterization (i.e., the $\mathrm{MPS}\_\sigma$ class), including those employing non-exponential activations (e.g., Absolute, Square, or Sigmoid). For these choices of $\sigma$, the classical Log-Sum-Exp (LSE) trick is not directly applicable. Only in the special case $\sigma=\exp$ does our method reduce to the standard LSE-based stabilization, which also explains its resemblance to techniques used in works such as EiNets that use the exponential function for positivity enforcement.
>
> Lastly, we have revised Section 3.5 to better illustrate the generality of the approach, by demonstrating its applicability to the Born Machine as well as added a paragraph clarifying the conditions under which LSF and LSE are equivalent and expanded on it in Appendix A.5
>
> ***W4: I would've like to see more of a comparison between the proposed approach and DMRF in terms of NLL, in addition to MNIST.***
>
> We have **added comparison against DMRG for all 20 density estimation** benchmarks (Table 4)

---

> ### Author Response · Authors · 2025-11-21
> **Response to questions (clarification of TN diagram and DMRG compatibility with AD)**
>
> ***Q1: What does line 4 in Figure 1 (a) correspond to? Does that denote matrix multiplication?***
>
> This is the TN diagram corresponding to a tensor contraction between two third order tensors, resulting in a fourth-order tensor. We added a pointer to a reference introducing TN diagrams in the caption of the Figure. ([https://tensornetwork.org/diagrams](https://tensornetwork.org/diagrams/))
>
> ***Q2: The authors mention that DMRG is non-differentiable. I expected that to translate to some form or biased relaxation, and therefore a higher NLL. However, that does not seem to be the case, at least on the MNIST experiments.***
>
> This is not precisely our claim. In our introduction we state that *DMRG is not fully compatible with automatic differentiation”*. We expand on this statement in Section 3.6 as well as provide a graphical example. Our argument is that integrating DMRG into modern machine learning frameworks is inherently difficult, as prior works relying on DMRG **do not define a differentiable loss function** (biased relaxation or otherwise) that enables end-to-end updates of all model parameters within an automatic-differentiation pipeline.

---

### Author Response · Authors · 2025-11-21
**General response to common concerns (novelty and addition of benchmarks)**

We would like to thank the reviewers for their thoughtful feedback. Based on comments of the reviewers we have made updates to the PDF file accordingly. All revisions are marked in blue text for clarity. We highlight below our response to questions shared by multiple reviewers:


***(1) The approach is straightforward with limited novelty***

We have revised our contribution statement in the introduction to more clearly articulate the scope of our contributions. Our contribution is the theoretical analysis of the instabilities that arise when training PTNs, together with experimental results demonstrating that a simple stabilization technique (LSF) can alleviate these instability issues while achieving performance competitive with DMRG on larger datasets. In summary, our contributions are:

1. **Theoretical analysis:** We provide a formal characterization of the numerical instabilities that arise when training PTNs such as $\text{MPS}\_\sigma$ and $\mathrm{MPS}\_{\mathrm{BM}}$. In particular, we prove lower bounds on the mean and variance of the computations associated with members of both classes, respectively.
2. **Experimental results:** We demonstrate empirically that, on real datasets, our stabilized training procedure achieves performance competitive with DMRG (a significantly more stable but computationally heavier algorithm than SGD) while running approximately **10× faster**.


***(2)  Is this just a direct application of Log-Sum-Exp trick?***

We would like to clarify that the stabilization procedure in Section 3.5 is *not* simply an application of the Log-Sum-Exp (LSE) trick. In particular, the LSE trick **cannot be directly employed for Born Machines**, since their probabilities are defined via squared amplitudes rather than through enforcing positivity by exponentiating individual MPS cores.

In addition to Born Machines, our stabilization procedure applies to any positive MPS parameterization (i.e., the $\mathrm{MPS}_\sigma$ class), including those employing non-exponential activations (e.g., Absolute, Square, or Sigmoid). For these choices of $\sigma$, the classical Log-Sum-Exp (LSE) trick is not directly applicable. Only in the special case $\sigma=\exp$ does our method reduce to the standard LSE-based stabilization, which also explains its resemblance to techniques used in works such as EiNets that use the exponential function for positivity enforcement.

Lastly, we have revised Section 3.5 to better illustrate the generality of the approach, by demonstrating its applicability to the Born Machine as well as added a paragraph clarifying the conditions under which LSF and LSE are equivalent and expanded on it in Appendix A.5


***(3) Can we compare our method LSF against DMRG on more benchmarks?***

We have updated Table 4, and now have a comparison between our method LSF and DMRG on **20 density estimation benchmarks**

---

### Meta-Review · Area_Chair_CQCk · 2025-12-28

**Summary:**

This paper proposes a numerically stable training procedure for probabilistic tensor networks based on logarithmic scale factors, enabling gradient-based learning of long MPS models within automatic differentiation frameworks. The submission was reviewed by four reviewers and the overall score was mixed to negative.

The reviewers consistently recognized the importance of addressing numerical instability in PTN training and acknowledged the practical effectiveness of the proposed approach. At the same time, several reviewers expressed concerns regarding the level of conceptual novelty and the clarity of the paper’s positioning with respect to existing work. While the rebuttal helped clarify certain aspects of the method, it did not fully alleviate these concerns, and no reviewer changed their initial score.

**Reviewer Concerns:**

The rebuttal and revisions helped address several concerns related to clarity and completeness, including improved explanation of the proposed stabilization method and additional experimental comparisons that strengthened the empirical evaluation. These changes improved the overall presentation and helped clarify the scope of applicability of the approach.

However, concerns regarding the level of conceptual novelty and the positioning of the method relative to existing stabilization techniques for tensor networks were only partially addressed. While the authors provided further discussion and clarification, one reviewer remained unconvinced that the contribution represents a sufficiently distinct advance beyond prior work.

**Reviewer Scores:**

The submission received four reviews with initial scores of 4, 4, 4, and 2. Based on the rebuttal and revisions, the additional clarifications and expanded experiments may have strengthened reviewers’ understanding of the work; however, the main concerns regarding novelty and positioning were only partially alleviated. As a result, it is likely that the three reviewers who initially gave a score of 4 would have maintained similar scores, while the reviewer who assigned a score of 2 would also likely have remained at that score.

---

### Decision · Program_Chairs · 2026-01-26

Reject